# Factors Contributing to Life-Change Adaptation in Family Caregivers of Community-Dwelling Individuals with Acquired Brain Injury

**DOI:** 10.3390/healthcare11192606

**Published:** 2023-09-22

**Authors:** Yuka Iwata, Etsuko Tadaka

**Affiliations:** 1Department of Community Health Nursing, Graduate School of Medicine, Yokohama City University, Yokohama 236-0004, Japan; 2Department of Community and Public Health Nursing, Graduate School of Health Sciences, Hokkaido University, Sapporo 060-0812, Japan

**Keywords:** acquired brain injury, caregiver, life-change adaptation

## Abstract

Acquired brain injury (ABI) is a public health issue that affects family caregivers, because individuals with ABI often require semi-permanent care and community support in daily living. Identifying the characteristics of family caregivers and individuals with ABI and examining life-change adaptation may provide valuable insights. The current study sought to explore the factors contributing to life-change adaptation in family caregivers of community-dwelling individuals with ABI. As a secondary analysis, a cross-sectional study was conducted using data obtained in a previous study of 1622 family caregivers in Japan. We hypothesized that life-change adaptation in family caregivers of individuals with ABI would also be related to family caregivers’ characteristics and the characteristics of individuals with ABI. In total, 312 valid responses were analyzed using Poisson regression analysis. The results revealed that life-change adaptation in family caregivers of individuals with ABI was related to sex (prevalence ratio [PR]: 0.65, confidence interval [CI]: −0.819;−0.041) and mental health (PR: 2.04, CI: 0.354; 1.070) as family caregivers’ characteristics, and topographical disorientation (PR: 1.51, CI: 0.017; 0.805) and loss of control over behavior (PR: 1.61, CI: 0.116; 0.830) as the characteristics of individuals with ABI, after adjusting for the effects of the caregiver’s age, sex, and the duration of the caregiver’s role. The current study expands existing knowledge and provides a deeper understanding to enhance the development of specific policies for improving caregiving services and supporting families.

## 1. Introduction

Cerebral injury from acquired brain injury (ABI) encompasses harm to the brain resulting from various causes, including traumatic incidents such as traumatic brain injury (TBI) or non-traumatic events, such as strokes, occurring after birth [1,2]. The repercussions of ABI often take the form of cognitive impairments in neurology, involving challenges such as compromised working memory, diminished executive function, delayed cognitive processing, and impaired attention. These effects can significantly impact a person’s autonomy in daily life [2,3,4,5]. Importantly, a substantial percentage of stroke patients [6] and TBI patients [7,8] meet the criteria for neurocognitive disorders outlined in the Diagnostic and Statistical Manual of Mental Disorders, Fifth edition (DSM-V).

In healthcare, ABI has emerged as a focal area [2,9]. In 2016, the global estimated prevalence of ABI exceeded 116.4 million individuals, reflecting a 3.6% upsurge since 1990 [10,11,12,13]. Despite this mounting epidemiological challenge, the management of ABI through scientifically validated neuroprotective agents for injury prevention or healing enhancement is currently lacking [2]. Moreover, the biological mechanism through which traumatic injury induces persistent cerebral damage, culminating in behavioral alterations, remains subject to study [14]. Individuals grappling with ABI often confront difficulties in executing daily tasks because of waning attention, concentration, and recognition capacities [1]. Meanwhile, in affected secondary neurodegeneration, impairments may manifest as post-traumatic epilepsy [15], sleep disturbances [16], and increased risk for suicide [17]. Within the context of ABI, these cognitive changes may either precede, succeed, or simultaneously occur along with behavioral changes [18]. Consequently, stakeholders involved in care of individuals with ABI must confront challenging, uncontrollable behavioral changes in their patients. These challenging conditions often necessitate semi-permanent care and communal support from families, with a significant proportion (approximately 92.3%) of ABI-affected individuals residing with their families [19].

The majority of family caregivers for those affected by ABI suddenly find themselves thrust into the role of caregiving across various stages of life. These caregivers typically find themselves in a pivotal phase of self-discovery, and research has extensively documented the adverse health consequences linked to caregiving. Prior research has predominantly focused on highlighting the adverse consequences encountered by family caregivers providing support to individuals affected by ABI. However, it is also important to understand the potential for favorable adaptations that may arise as a response to the transformative life shifts following the onset of ABI. Many caregivers find themselves undergoing a process of personal growth, wherein they uncover latent strengths and capacities while navigating uncharted territories. This transformative journey has the potential to exert a positive impact on the psychological and emotional well-being of family caregivers [13]. By cultivating empathetic aptitudes and bolstering stress-coping mechanisms, caregivers may forge deeper emotional connections and sustain a more constructive mental equilibrium.

Furthermore, it is imperative to recognize that the constructive changes experienced by family caregivers of ABI patients can also serve as a catalyst for the creation of novel social resources attuned to emerging healthcare needs. This process is significant not just for the caregivers in question, but also for society. The experiential insights gained by family caregivers represent a source of valuable information that could be useful for informing healthcare provisions and bolstering support initiatives. Consequently, care could be tailored to the needs of ABI patients and their families, potentially enhancing nationwide healthcare infrastructure in a holistic way. This paradigm may be useful for establishing a comprehensive and efficacious support framework that contributes not only to the well-being of family caregivers but also to improvement of the overall health of the community. We developed a reliable and valid instrument, the Life Change Adaptation Scale (LCAS), for family caregivers of individuals with ABI [13]. Using this instrument, future research can explore factors associated with this adaptation and advance the development of evidence-based intervention strategies. 

Previous studies attempted to predict family caregiver adaptive outcomes from the characteristics of family caregivers (e.g., sex [20], relationship [20,21], duration of care [22,23], mental health status [24,25,26]) or individuals with brain injury (e.g., sex [27], cognitive and personality changes, severity of injury [23]). The model of stress and coping among caregivers [28] is useful for enhancing and explaining life-change adaptation to stressors [29]. This theoretical model supports demographic factors as predisposing factors related to caregivers’ adaptations to stressors. Despite clear evidence that characteristics of family caregivers or individuals with brain injury generate adaptive outcomes for family caregivers cohabitating with community-dwelling individuals with ABI, there has been no rigorous factorial analysis using the LCAS or the model of stress and coping among caregivers. Identifying the characteristics of family caregivers and individuals with ABI, and life-change adaptations, deserves scientific recognition in terms of expanding our knowledge of LCAS, which has reliability and validity, or expanding the application of the model of stress and coping among caregivers. Moreover, in practical terms, identifying the characteristics of family caregivers and individuals with ABI may yield valuable information for designing interventions tailored to caregivers’ needs and enhance the development of specific policies for improving caregiving services and supporting families.

The International Classification of Functioning, Disability and Health (ICF), developed and proposed by the World Health Organization (WHO) [30], provides a comprehensive framework for understanding health and the determinants of individual and environmental factors. On the basis of this framework, we hypothesized that life-change adaptation of family caregivers of individuals with ABI is also related to the family caregiver’s individual characteristics, including age, age at the time of ABI occurrence, sex, relationship to the individual with ABI, and the characteristics of the individual with ABI, including age, age at the time of ABI occurrence, sex, relationship to their family caregiver, and the specific impairments resulting from ABI. However, the associations between life-change adaptation and the individual characteristics of family caregivers and individuals with ABI are currently unclear. 

Understanding the factors contributing to life-change adaptation in family caregivers of community-dwelling individuals with ABI has significant implications for healthcare. By investigating these factors, the current study addresses a critical aspect of caregiving that has far-reaching consequences. By examining both caregivers’ characteristics and the characteristics of individuals with ABI, the current study not only sheds light on the interplay between these factors but also has the potential to enhance current understanding of how to promote successful adaptation. From a public health perspective, this study could inform policy decisions by professionals such as public health nurses aiming to improve the overall health and quality of life of caregivers. Recognizing the significance of caregivers’ well-being in maintaining the health and stability of caregiving relationships could lead to the establishment of targeted support programs and resources. 

The purpose of the current study was to explore the factors contributing to life-change adaptation among family caregivers of community-dwelling individuals with ABI. We hypothesized that the characteristics of caregivers and individuals with ABI would be significant predictive factors for life-change adaptation status in family caregivers of individuals with ABI.

## 2. Materials and Methods

### 2.1. Data Source and Study Design

The current study was conducted as a secondary analysis using data from a previous study of 1622 members of the Japan ABI Family Caregivers Association between September 2019 and November 2020 [13]. The Japan ABI Family Caregivers Association is a representative organization for family caregivers of individuals with ABI in Japan. We used a maximum sample size that adhered to the criteria for the secondary analysis study, drawn from the total population of the primary study. Before sending the survey questionnaires, the author sent informed consent letters to the administrators of all associations of family caregivers of individuals with ABI. All 39 associations (47.5%) consented to participate in the previous study. The inclusion criteria in the current study were as follows: (1) the caregiver was caring for an individual with ABI, (2) the caregiver was aged 20 years or older, and (3) the individual with ABI developed ABI at 16 to 64 years old. The reason for including cases in which the individual with ABI developed ABI at 16 to 64 years old and excluding cases in which ABI occurred in an individual at <16 or >64 years old is that adult ABI does not typically progress with age, like pediatric ABI, and does not tend to exhibit decline with age, such as with ABI in old age [31]. Rather, adult ABI represents a stage that is associated with consistent cognitive and psychosocial function. A previous study did not support the notion that ABI inevitably causes ongoing neurological decline in the long-term post-injury period, because neurological impairment in old age includes decline associated with aging [32]. As a result, the adaptation process and outcomes for ABI are different to those of pediatric ABI and ABI in old age [13]. The study’s exclusion criteria were as follows: (1) missing data for the duration of the caregiver’s role, (2) missing data for the dependent variable, and (3) non-consent for data use. The sample size for the Poisson regression analysis was calculated using G*Power 3.1.2 on the basis of the assumptions of a two-tailed test, a moderate effect size of 0.15 (medium), and a significance level of 0.05. On the basis of a previous study [13], we expected the difference in prevalence between adaptive individuals and non-adaptive individuals to be 14.6%. Thus, the required sample size was calculated to be 161. The final sample size was 312, which exceeded the pre-study sample size calculation.

### 2.2. Measurements

#### 2.2.1. Dependent Variable

The dependent variable in this study was life-change adaptation, which was measured using the LCAS [13]. Life-change adaptation refers to “the outcome of adaptation to changes in living resources/health belief of life of family caregivers due to ABI” [13]. This scale consists of eight items. LCAS scores range from −24 to +24. The reliability of the scale has been established (Cronbach’s α: 0.84), and validity was confirmed for Japanese participants in a previous study [13]. Participants with LCAS scores of 0 or greater were categorized as the life-change adaptative group.

#### 2.2.2. Independent Variables

On the basis of the International Classification of Functioning, Disability and Health (ICF), we included characteristics of family caregivers of individuals with ABI as individual factors, such as the caregiver’s own age, age at the time of ABI occurrence, sex, relationship to an individual with ABI, duration of the caregiver’s role, and mental health. Additionally, we included characteristics of individuals with ABI as environmental factors, such as age, age at the time of ABI occurrence, and sex of individuals with ABI who were receiving care, as well as various impairments.

##### Characteristics of Family Caregivers of Individuals with ABI 

Participants’ characteristics included age (<65 years old vs. ≥65 years old), age at the time of ABI occurrence (<50 years old vs. ≥50 years old), sex (female vs. male), relationship to an individual with ABI (parent vs. spouse and other), duration of a caregiver’s role (<120 months vs. ≥120 months), and mental health (mood or anxiety disorders vs. no mood or anxiety disorders). The Japanese version of the Kessler-6 (K6) [33] was used to measure mental health. This scale consists of six items. K6 scores ranged from 0 to 24. High scores on the K6 indicate a low level of mental health. The reliability of the scale has been established (Cronbach’s α: 0.88). In this scale, the area under the receiver operating characteristic curve indicated excellent screening ability for the Diagnostic and Statistical Manual of Mental Disorders, Fourth edition (DSM-IV) classification of mood and anxiety disorders [33]. A previous study supports the high performance of the Japanese version of the K6 in screening for mood and anxiety disorders with optimal cut-off points of 4/5 [34]. Thus, participants with a K6 score of 5 or greater were categorized as the mood or anxiety disorders group.

##### Characteristics of Individuals with ABI

The examined characteristics of individuals with ABI included age (<50 years old vs. ≥50 years old), age at the time of ABI occurrence (<35 years old vs. ≥35 years old), and sex (female vs. male). Utilizing the key domains of cognition defined in the DSM-5 [35], the following impairments were also included: impaired complex attention vs. no impaired complex attention; poor executive function vs. no poor executive function; loss of learning and memory vs. no loss of learning and memory; aphasia vs. no aphasia; deficits of spatial awareness vs. no deficits of spatial awareness; deficits of physical awareness vs. no deficits of physical awareness; deficits of disease awareness vs. no deficits of disease awareness; apraxia vs. no apraxia; topographical disorientation vs. no topographical disorientation; and loss of control over behavior vs. no loss of control over behavior. Additionally, the cause of ABI (trauma to the head or stroke or tumor or anoxia or infection or other) was included to examine generalizability in this study.

### 2.3. Statistical Analysis

Chi-squared analyses and the Mann–Whitney U test were used to examine whether there were differences between life-change adaptative and non-adaptative family caregivers in terms of demographics. The distributions of the quantitative variables were verified using the Lilliefors-corrected Kolmogorov–Smirnov normality test with *p* < 0.05 for all quantitative variables. After identifying significant variables from the Chi-squared analyses and using the Mann–Whitney U test to examine whether there were differences between life-change adaptive and non-adaptive family caregivers in terms of variables that were adjusted for age and sex and duration of a caregiver’s role, we conducted the Poisson regression analyses. There were no missing data for the dependent variable. The covariate and independent variables had missing data. Considering the potential for imputing missing data to introduce distortions in the results and recognizing that the influence of missing data on the overall conclusions was limited, we proceeded with the analysis without data imputation. In the Poisson regression analysis, adaptive participants with LCAS scores of 0 to 24 were classified as 1, and non-adaptive participants with LCAS scores of −24 to −1 were classified as 0. Using the LCAS, a score of 0 or a positive value indicated that family caregivers of individuals with ABI assessed their life change caused by ABI as a strengthening life event, and negative scores indicated that family caregivers of individuals with ABI assessed their life change caused by ABI as a problem [13]. A *p*-value < 0.05 or a 95% confidence interval (CI) that did not include 1 indicated that the results were statistically significant. The data were analyzed using the SPSS software package (version 28.0; IBM Corp., Armonk, NY, USA).

### 2.4. Ethical Approval

This research was conducted in accordance with the 1964 Declaration of Helsinki (and its amendments) and the ethical guidelines for life sciences and medical research involving human subjects presented by the Ministry of Health, Labour and Welfare of Japan. The Institutional Review Board of the Medical Department of Yokohama City University School approved this study on 17 July 2019 (No. A190700007).

## 3. Results

### 3.1. Characteristics of the Sample

A total of 312 individuals were included in this study. Participants ranged in age from 31 to 83 years (mean age 62.8 years, SD = 10.3 years). A total of 76.0% of family caregivers were female. The duration of the caregiver’s role ranged from 6 months to 40 years (mean 151.7 months, SD = 96.8). Individuals with ABI ranged in age from 21 to 77 years (mean age 49.4 years, SD = 12.2). A total of 81.3% of individuals with ABI were male. ABI was caused by trauma to the head in 50.3% of individuals, by stroke in 43.6%, by anoxia in 5.1%, by infection in 4.5%, and by tumor in 4.2%.

### 3.2. Comparison of Life-Change Adaptation Status

Table 1 indicates the results of life-change adaptation status comparisons for study variables. As shown in Table 1, there were significant differences between the life-change adaptive and non-adaptive groups in terms of caregiver’s age (*p* < 0.01), caregiver’s age at the time of ABI occurrence (*p* < 0.01), caregiver’s sex (*p* < 0.01), relationship to individuals with ABI (*p* < 0.05), mental health (*p* < 0.01), individuals with ABI’s sex (*p* < 0.05), poor executive function (*p* < 0.01), loss of learning and memory (*p* < 0.01), deficits of spatial awareness (*p* < 0.05), deficits of physical awareness (*p* < 0.05), deficits of disease awareness (*p* < 0.01), topographical disorientation (*p* < 0.01), and loss of control over behavior (*p* < 0.01). 

Considering the above results and taking multicollinearity into consideration, we included eight of the thirteen factors as independent variables, and age, sex, and duration of the caregiver’s role as control variables in the Poisson regression analysis. Table 2 shows related factors in the Poisson regression of life-change adaptation status. Factors significantly associated with life-change adaptation status were caregiver’s sex (PR: 0.65, CI: −0.819; −0.041), mental health (PR: 2.04, CI: 0.354; 1.070), topographical disorientation (PR: 1.51, CI: 0.017; 0.805), and loss of control over behavior of individuals with ABI (PR: 1.61, CI: 0.116; 0.830). 

## 4. Discussion

In the current study, we investigated the relationship between demographic factors and life-change adaptation in a representative population of family caregivers in Japan. While factors influencing life-change adaptation in family caregivers of individuals with ABI remain controversial [13,20,24,25,27,29], the current study provided a novel investigation of life-change adaptation of family caregivers cohabitating with community-dwelling ABI patients, exploring the respective factors for both patients and family caregivers, using reliable and valid measures. Our study aimed to contribute to the discovery of new knowledge by conducting a comprehensive and rigorous analysis of life-change adaptation in family caregivers cohabitating with community-dwelling individuals with ABI. The current findings warrant further examination of the LCAS and an expansion of the application of the model of stress and coping among caregivers. The current findings may inform the development of specific policies for improving caregiving services and supporting families. Furthermore, the current study employed a robust methodology and a representative sample, enhancing the reliability and validity of the results. This not only provides new insights to extend current knowledge, but it also provides a crucial foundation for future research. By understanding the factors influencing life-change adaptation in family caregivers cohabitating with community-dwelling individuals with ABI, the current findings may aid in the provision of more appropriate support and resources, thereby improving responses to related issues. Regarding the demographics of family caregivers of individuals with ABI, the caregivers were mostly women (76.0%). The average age of individuals with ABI at the time of ABI occurrence was 49.4 years (SD = 12.2). According to the official evaluation by the Japanese government and a previous study, this is similar to the demographic profile of participants in the survey of family caregivers of individuals with ABI [19]. Thus, the sample was deemed to be representative of the population of family caregivers of individuals with ABI.

In the current study, the key findings of the Poisson regression analysis indicated that the loss of control over behavior (PR: 1.61, CI: 0.116; 0.830) and topographical disorientation (PR: 1.51, CI: 0.017; 0.805) of individuals with ABI significantly predicted life-change adaptation in family caregivers of individuals with ABI, after controlling for the caregiver’s age, sex, and duration of caregiving. A previous study reported that family caregivers of individuals with ABI with impaired behavioral control, as well as cognitive and memory impairments, face higher levels of increased care burden propensity compared with family caregivers of individuals with ABI without impaired behavioral control [36]. However, in contrast to this prior study [36], only loss of control over behavior and topographical disorientation were found to exhibit a significant association with life-change adaptation in family caregivers in the current study. Hence, in considering life-change adaptation interventions, it may be beneficial to adopt existing programs to manage impairments [37,38]. There is evidence that loss of control over behavior affects family cohesion [39,40]. Family cohesion and good social support are reported to be predictive of better adaptation of family caregivers after brain injury [41]. Moreover, there is evidence that becoming lost as a result of topographical disorientation causes psychological distress in both patients and caregivers, and it increases the odds of being institutionalized [42]. Hence, in considering life-change adaptation interventions, it is necessary to assess and identify the specific characteristics of not only the dyad of the caregiver and the individual with ABI, but also the family as a whole, and to consider a care program according to these characteristics. Additionally, the majority of previous studies attempted to predict family caregiver adaptation from characteristics of individuals with brain injury (e.g., cognitive and personality changes, duration of time since injury, and severity of injury) [23]. However, according to the cognitive model, it is not the objective characteristics of the caregiving situation that determine adaptation, but the caregiver’s appraisal of their situation [43]. Hence, in considering life-change adaptation interventions, it is necessary to investigate in detail the specific characteristics of loss of control over behavior that affect the caregiver’s appraisal and to consider care programs according to these characteristics.

The results revealed that mental health (PR: 2.04, CI: 0.354; 1.070) was significantly associated with life-change adaptation. A previous investigation postulated that mental health could potentially exert an influence on life-change adaptation [13]. Furthermore, another study reported that the observed linkage between mental health and adaptation circumstances can be elucidated through the lens of the model of stress and coping among caregivers [28]. Both theoretical and empirical research has indicated that addressing an individual’s needs can help manage stress and promote better adaptation to brain injury-related difficulties [44]. Given the challenges of coping with changes resulting from acquired injuries, family caregivers of individuals with ABI are frequently confronted with stressors from unmet needs [45]. This predicament may pose a difficult challenge in the process of life-change adaptation.

The results revealed that a caregiver’s sex (PR: 0.65, CI: −0.819; −0.041) was significantly associated with life-change adaptation. A previous study suggested that a caregiver’s sex may impact psychological adaptation [20]. There is consistent evidence that male caregivers perform better in terms of adaptive status compared with female caregivers, as follows. First, males and females tend to have different coping styles, with females taking on caregiving roles with more negative appraisal. In situations involving strong stressors related to caregiving, coping with a stressor using positive appraisal facilitates adaptation to stressors [28,29]. Several previous studies suggest that male caregivers exhibit a reduced frequency of negative appraisals of caregiving compared with their female counterparts [20,46,47,48]. The second finding is that sex differences exist between family caregivers and individuals with ABI. In the current study, a total of 76.0% of family caregivers were female, and 81.3% of individuals with ABI were male. Additionally, reduced personal sexual appeal has been reported as a common personality change following brain injury [49,50], which may affect pair-bonding between female family caregivers and men with ABI [51]. To cope with acquired injury changes, female caregivers are often asked to renegotiate relationships [52], which may be challenging for life-change adaptation.

The current study involved several limitations that should be considered. First, because this was a cross-sectional study, causal relationships may not be inferred. Second, the associations between study variables might be explained by one or more unmeasured confounding variables (e.g., comorbidities). Third, this study was a secondary analysis using data from a study of 1622 members of the Japan ABI Family Caregivers Association. The Japan ABI Family Caregivers Association is a representative organization for family caregivers of individuals with ABI in Japan. However, it is important to note that this study did not include individuals who are not affiliated with the Japan ABI Family Caregivers Association, and participants were not randomly selected. Fourth, family caregivers who were likely to have a lower level of life-change adaptation (e.g., those who had given up caregiving responsibilities because they were growing older) were not included in this study. Thus, survival bias may have influenced the observed factors. Finally, study participants’ ages ranged from 31 to 83 years old, with an average age of 62.8 years. Although caregivers may have difficulty adapting to new life situations or other life changes as they age, life-change adaptation to aging in caregivers was outside of the scope of the current study. Despite these limitations, this study, which provides a psychological and sociological examination of ABI, can deepen the current understanding of the diverse factors that affect family caregivers of individuals with ABI via the integration of foundational research that explores biological mechanisms through which traumatic injuries lead to persistent cerebral damage [14] or comprehends secondary neurodegeneration in ABI [18] with the applied research that highlights their relevance to caregivers. This fusion of fundamental and applied methods can contribute to the development of a comprehensive approach to improve caregiving for individuals with ABI.

## 5. Conclusions

This study explored the factors contributing to life-change adaptation in family caregivers of community-dwelling individuals with ABI and explored respective factors for both patients and family caregivers using reliable and valid measures. Importantly, the current findings revealed that life-change adaptation in family caregivers of individuals with ABI may also be related to sex and mental health of family caregivers, as family caregivers’ characteristics, and to topographical disorientation and loss of control over behavior of individuals with ABI, as characteristics of individuals with ABI. The current findings contribute to the expansion of existing knowledge, providing a deeper understanding of this issue and contributing to the development of specific policies for improving caregiving services and supporting families.

## Figures and Tables

**Table 1 healthcare-11-02606-t001:** Comparison of evaluation measures between adaptive and non-adaptive groups.

n = 312					
		Life-Change Adaptation Status		*X*^2^ or Mann–Whitney U Test	*p*-Value
		Mean (SD) or %			
		Adaptive n = 136	Non-Adaptive n = 176		
Life-change adaptation (score)		4.2 (4.4)	−7.4 (5.5)	-	-
Caregiver’s age (years)		64.8 (10.0)	61.1 (10.2)	−3.2	0.003 **
(n = 299)	<65 years old	43.4	56.8	5.6	0.012 *
	≥65 years old	52.2	39.2		
	Missing	4.4	4.0		
Caregiver’s age at the time of ABI occurrence		53.0 (10.4)	54.8 (73.3)	−3.0	0.009 **
(n = 298)	<50 years old	32.4	40.9	2.2	0.085
	≥50 years old	62.5	55.1		
	Missing	5.1	4.0		
Caregiver’s sex					
(n = 306)	Female	68.4	80.7	6.2	0.014 *
	Male	29.4	17.6		
	Missing	2.2	1.7		
Relationship to individual with ABI					
(n = 309)	Parent	60.3	46.0	6.3	0.043 *
	Spouse and other	39.0	52.8		
	Missing	0.7	1.1		
Duration of a caregiver’s role (months)		152.7 (100.6)	150.0 (94.4)	−0.3	0.717
(n = 312)	<120 months	44.1	41.5	0.2	0.362
	≥120 months	55.9	58.5		
Mental health (score) ^a^		5.9 (4.9)	9.8 (5.6)	6.1	<0.001 **
(n = 302)	<5 points	45.6	18.8	28.8	<0.001 **
	≥5 points	49.3	79.5		
	Missing	5.1	1.7		
Age of individuals with ABI		48.4 (11.7)	50.1 (12.4)	1.2	0.178
(n = 311)	<50 years old	58.8	47.7	4.1	0.028 *
	≥50 years old	40.4	52.3		
	Missing	0.7	0.0		
Age of individuals with ABI at the time of ABI occurrence		36.4 (13.6)	38.0 (14.7)	1.0	0.379
(n = 312)	<35 years old	46.3	44.9	0.1	0.445
	≥35 years old	53.7	55.1		
Sex of individuals with ABI					
(n = 311)	Female	25.0	14.8	5.1	0.018 *
	Male	75.0	84.7		
	Missing	0.0	0.6		
Impairment					
(n = 311)	Impaired complex attention	88.2	90.3	0.2	0.407
	Poor executive function	83.1	92.6	6.1	0.011 *
	Loss of learning and memory	75.0	99.4	7.4	0.005 **
	Aphasia	35.3	44.9	2.8	0.061
	Deficits of spatial awareness	17.6	27.8	4.3	0.025 *
	Deficits of physical awareness	8.1	15.3	3.6	0.041 *
	Deficits of disease awareness	5.9	15.3	6.7	0.007 **
	Apraxia	22.1	30.1	2.3	0.081
	Topographical disorientation	29.4	44.9	7.3	0.005 **
	Loss of control over behavior	48.5	69.9	13.6	<0.001 **

* *p* < 0.05, ** *p* < 0.01 (two-sided). ^a^ Mental health: The score of the Japanese version of the K6 [33].

**Table 2 healthcare-11-02606-t002:** Factors associated with life-change adaptation in family caregivers.

				n = 312
Variables	PR	95% CI		*p*-Value
		Low	High	
Caregiver’s age	1.30	−0.095	0.616	0.151
Caregiver’s sex	0.65	−0.819	−0.041	0.030 *
Duration of the caregiver’s role	0.92	−0.435	0.279	0.668
Relationship to individual with ABI	1.21	−0.235	0.615	0.381
Mental health	2.04	0.354	1.070	<0.001 **
Sex of individuals with ABI	0.89	−0.561	0.326	0.604
Loss of learning and memory	1.39	−0.093	0.750	0.126
Deficits of spatial awareness	1.53	−0.054	0.906	0.082
Deficits of disease awareness	1.82	−0.120	1.318	0.102
Topographical disorientation	1.51	0.017	0.805	0.041 *
Loss of control over behavior	1.61	0.116	0.830	0.009 **

PR: prevalence ratio; CI: confidence intervals; *: *p* < 0.05; **: *p* < 0.01. Life-change adaptation (1: adaptive 0: non-adaptive); caregiver’s age (1: ≥65 years old 0: <65 years old); caregiver’s sex (1: female 0: male); duration of the caregiver’s role (1: ≥120 months 0: <120 months); relationship to the individual with ABI (1: spouse and other 0: parent); mental health (1: mood or anxiety disorders 0: not individuals with mood or anxiety disorders); sex of individuals with ABI (1: female 0: male); loss of learning and memory (1: yes 0: no); deficits of spatial awareness (1: yes 0: no); deficits of disease awareness (1: yes 0: no); topographical disorientation (1: yes 0: no); loss of control over behavior (1: yes 0: no). Missing data were excluded.

## Data Availability

The data that support the findings of this study are available from Yokohama City Local Government and Yokohama City University, but restrictions apply to the availability of these data under the Japan Personal Information Protection Law, since they were used under license for the current study, and so are not publicly available. Data are however available from the first/corresponding authors upon reasonable request and with permission of Yokohama City Local Government and Yokohama City University.

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
