# Peer review of "Factors Contributing to Life-Change Adaptation in Family Caregivers of Community-Dwelling Individuals with Acquired Brain Injury"

_healthcare, 2023, doi:10.3390/healthcare11192606_

Round 1

Reviewer 1 Report

This manuscript significantly interests healthcare readers because of the public health importance of ABI. Nevertheless, it is imperative to address certain concerns, such as providing additional information to facilitate the study’s replication by other researchers.

Mayor changes:

a) In this cross-sectional study would be better performance prevalence ratio instead of odds ratio.

Please, the authors can see the following references:

Espelt A., Marí-Dell’Olmo M., Penelo E., Bosque-Prous M. Applied Prevalence Ratio estimation with different Regression models: An example from a cross-national study on substance use research. Adicciones. 2016;29:105–112. doi: 10.20882/adicciones.823.

Barros A.J.D., Hirakata V.N. Alternatives for logistic regression in cross-sectional studies: An empirical comparison of models that directly estimate the prevalence ratio. BMC Med. Res. Methodol. 2003; 3:21. doi: 10.1186/1471-2288-3-21

b) The introduction should be a precise overview of the current state of research, and it should be immediately connected with your research problem. The authors should include information about the theoretical scope or the framework of the topic researcher and they did not justify the interest of their study and how your research will contribute to the existing research. In addition, the information of the questionnaire developed to assess Life Change Adaptation Scale is unnecessary. Thus, it is suggested to the authors to rewrite again the Introduction.

c) The authors should include information to explain they did not use all the sample size in this sub study. In addition, they include information about the sample size calculation.

d) How were the missing values taken into account in the analysis?

e) In the table 2 should eliminate de Beta results and provide the p-value of the results. In addition, they should justify as the dependent variables were selected.

Minor changes

a) The symbol “±” used to describe the results should be avoided. It is preferable to present the measures of variability in brackets or as confidence intervals (please, the authors can see the following references).

1. Altman DG, Gore SM, Gadner MJ, Pocock SJ. Statistical guidelines for contributors to medical journals. Br Med J 1983; 286: 1.489-1.493
2. Bailar JC, Mosteller F. Guidelines for statistical reporting in articles for medical journals: amplifications and explanations. Ann Intern Med 1988; 108: 266-273

b) Include more detailed information of descriptive analysis in the statistical analysis, for example how to check the normality of the continuous variables.

c) The first paragraph of the results should summarise the major results of the study.

d) Please provide the strength of the study in the discussion. Discuss the generalisability (external validity) of the study results

e) The conclusion of the study should include to the answer of research question. Please, revise.

Author Response

To the comments of Reviewer 1

 Major changes:

a) In this cross-sectional study would be better performance prevalence ratio instead of odds ratio. Please, the authors can see the following references: Espelt A., Marí-Dell’Olmo M., Penelo E., Bosque-Prous M. Applied Prevalence Ratio estimation with different Regression models: An example from a cross-national study on substance use research. Adicciones. 2016;29:105–112. doi: 10.20882/adicciones.823.; Barros A.J.D., Hirakata V.N. Alternatives for logistic regression in cross-sectional studies: An empirical comparison of models that directly estimate the prevalence ratio. BMC Med. Res. Methodol. 2003; 3:21. doi: 10.1186/1471-2288-3-21

Response: We agree the reviewer’s suggestion. We performed a Poisson regression analysis with the life change adaptation/non-adaptation group as the dependent variable and the other variables as independent variables. The Prevalence Ratio and 95% Confidence Interval of life change non-adaptation were calculated for each independent variable, and the table was revised. We have revised the analysis explanation in this context, the result, discussion and table 2.

Line. 19-26

“In total, 312 valid responses and 285 individuals were analyzed for the Poisson regression analysis. The results revealed that life-change adaptation in family caregivers of individuals with ABI would also be related to sex (prevalence ratio [PR]: 0.65; confidence interval [CI]: -0.819 - -0.041), mental health (PR: 2.04; CI: 0.354 - 1.070) as family caregivers’ characteristics and topographical disorientation (PR: 1.51; CI: 0.017 - 0.805) and loss of control over behavior of individuals with ABI (PR: 1.61; CI: 0.116 - 0.830) as the characteristics of individuals with ABI after adjusting for the ef-fects of caregiver’s age, sex, and the duration of the caregiver’s role.”

Line. 200-204

“After identifying significant variables from the Chi-squared analyses and the Mann-Whitney U test were used to examine whether there were differences between life-change adaptive and non-adaptive family caregivers in terms of variables adjusting for age and sex and duration of a caregiver’s role, we conducted the Poisson regression analyses.”

b) The introduction should be a precise overview of the current state of research, and it should be immediately connected with your research problem. The authors should include information about the theoretical scope or the framework of the topic researcher and they did not justify the interest of their study and how your research will contribute to the existing research. In addition, the information of the questionnaire developed to assess Life Change Adaptation Scale is unnecessary. Thus, it is suggested to the authors to rewrite again the Introduction.

Response: We appreciate the reviewer’s suggestion. The introduction was corrected to include information on the theoretical scope and topic framework, and to justify the research interest and how the study contributes to existing research in the introduction. We also removed unnecessary information. We have added the following text:

Line. 82-92

“The model of stress and coping among caregivers [23] is a useful model to enhance and explain life-change adaptation to stressors [24]. This theoretical model supports demographic factors as predisposing factors related to caregivers’ adaptation to stressors. Despite clear evidence that characteristics of family caregivers or individuals with brain injury generate adaptive outcomes for family caregivers cohabitating with community-dwelling individuals with ABI, there has been no rigorous factorial analysis using the LCAS or the model of stress and coping among caregivers, which has reliability and validity. Identifying the characteristics of family caregivers and individuals with ABI, and life-change adaptation, may deserves scientific recognition in term of expanding our knowledge of LCAS which has reliability and validity or expanding the application of the model of stress and coping among caregivers.”

c) The authors should include information to explain they did not use all the sample size in this sub study. In addition, they include information about the sample size calculation.

Response: We appreciate the reviewer’s suggestion. We have added the following text:

Line. 146-150

“The sample size for Poisson regression analysis was calculated using G*Power 3.1.2 based on the assumptions of a two-tailed test, a moderate effect size of 0.15 (medium), a significance level of 0.05. Based on a previous study [13], we expected a difference between an adaptive prevalence and non-adaptive of 14.6%. Thus, the required sample size was 161.”

d) How were the missing values taken into account in the analysis?

Response: We appreciate the reviewer’s suggestion. We have added the following text:

Line. 204-209

“There was no missing data in dependent variable. The covariate and independent variables had missing data. Considering the potential for imputing missing data to introduce result distortions, and recognizing that the influence of missing data on the overall conclusions was limited, we proceeded with the analysis without data imputation. Thus, all 285 data were used for the Poisson regression analyses.”

e) In the table 2 should eliminate de Beta results and provide the p-value of the results. In addition, they should justify as the dependent variables were selected.

Response: We agree the reviewer’s suggestion. We have revised table 2. Also, we added the information both of why the independent variables selected (Line. 161-167) and why the dependent variable was selected (Line. 209-214).

Line. 161-167

“Based on the theory of the International Classification of Functioning, Disability and Health (ICF), we included characteristics of family caregivers of individuals with ABI as an individual factor such as the caregiver's own age, age at the time of ABI occurrence, sex, relationship to an individual with ABI, duration of the caregiver's role, and mental health. Additionally, we considered included characteristics of individuals with ABI as an environmental factor such as the age, age at the time of ABI occurrence, and sex of the individuals with ABI who were receiving care, as well as various impairment.”

Line. 209-214

“In the Poisson regression analysis, adaptive group at the LCAS score 0 to 24 classified as 1 and non-adaptive group at the LCAS score -24 to -1 classified as 0. Using the LCAS, with 0 and positive score indicating that a family caregiver of individuals with ABI assessed their life change cause ABI as strength life event, and With negative score indicating that a family caregiver of individuals with ABI assessed their life change cause ABI as problem[13].”

Minor changes:

a) The symbol “±” used to describe the results should be avoided. It is preferable to present the measures of variability in brackets or as confidence intervals (please, the authors can see the following references). 1. Altman DG, Gore SM, Gadner MJ, Pocock SJ. Statistical guidelines for contributors to medical journals. Br Med J 1983; 286: 1.489-1.493; 2. Bailar JC, Mosteller F. Guidelines for statistical reporting in articles for medical journals: amplifications and explanations. Ann Intern Med 1988; 108: 266-273

Response: We appreciate the reviewer’s suggestion. As suggested, we corrected the standard deviation of the measures is indicated in brackets in Table 1.

b) Include more detailed information of descriptive analysis in the statistical analysis, for example how to check the normality of the continuous variables.

Response: We appreciate the reviewer’s suggestion. We have added the following text:

Line. 197-214

“Chi-squared analyses and the Mann-Whitney U test were used to examine whether there were differences between life-change adaptative and non-adaptative family caregivers in terms of demographics. The distributions of the quantitative variables were verified by the Shapiro-Wilk normality test with p<0.05 for all quantitative variables. After identifying significant variables from the Chi-squared analyses and the Mann-Whitney U test were used to examine whether there were differences between life-change adaptive and non-adaptive family caregivers in terms of variables adjusting for age and sex and duration of a caregiver’s role, we conducted the Poisson regression analyses. There was no missing data in dependent variable. The covariate and independent variables had missing data. Considering the potential for imputing missing data to introduce result distortions, and recognizing that the influence of missing data on the overall conclusions was limited, we proceeded with the analysis without data imputation. Thus, all 285 data were used for the Poisson regression analyses. In the Poisson regression analysis, adaptive group at the LCAS score 0 to 24 classified as 1 and non-adaptive group at the LCAS score -24 to -1 classified as 0.”

c) The first paragraph of the results should summarize the major results of the study.

Response: We appreciate the reviewer’s suggestion. We have added the following text:

Line. 225-230

“In total, 312 valid responses were analyzed. The results revealed that caregiver’s sex (prevalence ratio [PR]: 0.65; CI: -0.819 - -0.041), mental health (PR: 2.04; CI: 0.354 - 1.070), topographical disorientation (PR: 1.51; CI: 0.017 - 0.805) and loss of control over behavior of individuals with ABI (PR: 1.61; CI: 0.116 - 0.830) were significantly associated with life-change adaptation after adjusting for the effects of caregiver’s age, sex, and the duration of the caregiver’s role.”

d) Please provide the strength of the study in the discussion. Discuss the generalisability (external validity) of the study results.

Response: We appreciate the reviewer’s suggestion. We have added the following text:

Line. 274-283

“the strength of the current study that providing a novel investigation of life-change adaptation of family caregivers cohabitating with community-dwelling ABI patients, exploring the respective factors for both patients and family caregivers, using reliable and valid measures. Our study aimed to contribute to the discovery of new knowledge by conducting a comprehensive and rigorous analysis of life-change adaptation in family caregivers cohabitating with community-dwelling individuals with ABI. The current findings deserve expanding our knowledge of LCAS and expanding the application of the model of stress and coping among caregivers. The current findings may inform the development of specific policies for improving caregiving services and supporting families.”

Line. 289-295

“Regarding the demographics of the family caregivers of individuals with ABI, the caregivers were mostly women (76.0%). The average age of the individual with ABI at the time of ABI occurrence was 49.4 years (SD = 12.2). According to the official evaluation by the Japanese government and a previous study, this is nearly identical to the profile of participants in the survey on the family caregivers of individuals with ABI [14]. Thus, the sample was deemed representative of the population of family caregivers of individuals with ABI.”

e) The conclusion of the study should include to the answer of research question. Please, revise.

Response: We appreciate the reviewer’s suggestion. We have added the following text:

Line366-369

“The findings of the study unveiled a noteworthy that life-change adaptation in family caregivers of individuals with ABI would also be related to sex and mental health as family caregivers’ characteristics and topographical disorientation and loss of control over behavior as the characteristics of individuals with ABI.”

Reviewer 2 Report

Dear authors, thank you for allowing me to review your work titled “Factors contributing to life-change adaptation in family care-givers of community-dwelling individuals with acquired brain injury”.

I see the manuscript as a cross-sectional study using data from a previous study (I suppose it is the manuscript reference 27).

From my point of view, the strong point is being a relevant issue and the methodology (except the sample).

However, there are some flaws to be considered previous its publicación.

Majors considerations,

Materials and Methods

The main one is, from my perspective, the sample. Authors mention, in lines 216 and 227-228 a “large representative sample”, when the sample is 312 individuals. Because of the population of Japan, why do authors affirm that is a representative sample?

Authors should concrete in “Materials and Methods” how do they calculate the sample, and if it is a random sample or a convenience one, arguing it in the “Discussion” section as a limitation in case it is not a random sample. 

Results

In Table 1, I can´t see neither in “mental health” nor in “impairment” the analysis comparing sex (female, male). 

Conclusions,

I suggest to be re-written, because authors just describe the study and the title. Please, concrete what relevant “findings contribute to the expansión of existing knowledge, providing a deeper understanding of this issue and contributing to the development of specific policies…”, and what predictive factors contribute to life-change adaptations.

Minors considerations,

Lines 52-53. Please, specify which “such positive life-change adaptations” that can be beneficial for caregivers.

And in line 93, how to promote successful interventions. What kind of interventions?

Lines 113-128. Why ABI in old age decline with age. Please include some external reference which evidence this statement.

In line 118 authors mention that “adult ABI represents a stage that is associated with consistent cognitive and psychosocial function”. But, why is consistent? If every person is different and the psychosocial enviroment could be different too…  

Lines 258-259, about male caregivers perform statement, please support it with a reference evidence.

In limitations, please, consider the “survival bias” given the age of the individuals in the study.

In lines 276-278, and accordingly to the manuscript title, why life-change adaptation to aging in care givers was outside the scope of the currently study?.

An asterisc is missed en table 1 “Deficits of spatial awareness… 0.041*

Author Response

To the comments of Reviewer 2

  1. The main one is, from my perspective, the sample. Authors mention, in lines 216 and 227-228 a “large representative sample”, when the sample is 312 individuals. Because of the population of Japan, why do authors affirm that is a representative sample?

Response: We appreciate the reviewer’s suggestion. As pointed out, the generalizability is indicated and the term "large (sample)" has been removed. We have added the following text:

Line. 289-295

“Regarding the demographics of the family caregivers of individuals with ABI, the caregivers were mostly women (76.0%). The average age of the individual with ABI at the time of ABI occurrence was 49.4 years (SD = 12.2). According to the official evaluation by the Japanese government and a previous study, this is nearly identical to the profile of participants in the survey on the family caregivers of individuals with ABI [14]. Thus, the sample was deemed representative of the population of family caregivers of individuals with ABI.”

  1. Authors should concrete in “Materials and Methods” how do they calculate the sample, and if it is a random sample or a convenience one, arguing it in the “Discussion” section as a limitation in case it is not a random sample.

Response: We appreciate the reviewer’s suggestion. As pointed out, the generalizability is indicated and the term "large (sample)" has been removed. We have added the following text:

Line. 126-129

“The current study was conducted as a secondary analysis of convenience sampling data from a previous study, obtained from 1,622 family caregivers of individuals with ABI who belonged to associations for families in Japan and were selected as eligible participants between September 2019 and November 2020 [13].”

Line. 353-356

“Third, a convenience sampling of institutions in Japan, rather than probability sampling was used. In generalizing the findings of this study, one must consider that the study was conducted through institutions of family caregivers in Japan.”

  1. In Table 1, I can´t see neither in “mental health” nor in “impairment” the analysis comparing sex (female, male).

Response: Analyses identifying differences in mental health or impairment by gender group are outside the purpose of this study and therefore are not described.

  1. Conclusions, I suggest to be re-written, because authors just describe the study and the title. Please, concrete what relevant “findings contribute to the expansión of existing knowledge, providing a deeper understanding of this issue and contributing to the development of specific policies…”, and what predictive factors contribute to life-change adaptations.

Response: We appreciate the reviewer’s suggestion. We have corrected the conclusion and added the following text:

Line. 366-370

“The findings of the study unveiled a noteworthy that life-change adaptation in family caregivers of individuals with ABI would also be related to sex and mental health as family caregivers’ characteristics and topographical disorientation and loss of control over behavior as the characteristics of individuals with ABI.”

  1. Lines 52-53. Please, specify which “such positive life-change adaptations” that can be beneficial for caregivers. And in line 93, how to promote successful interventions. What kind of interventions?

Response: We appreciate the reviewer’s suggestion. We have described specifics of “such positive life-change adaptations” that can be beneficial for caregivers. We cannot indicate in this study paper how to promote the success of the intervention, as what kind of intervention is a topic for future research. However, we have specified who will provide the intervention, because public health nurses are the key providers of the intervention to family caregiver of individuals with ABI in Japan. We have corrected the following text:

Line. 57-63

“Many caregivers find themselves undergoing a process of personal growth, wherein they uncover latent strengths and capacities while navigating uncharted territories. This transformative journey bears the potential to wield a positive impact on the psychological and emotional well-being of family caregivers [13]. By cultivating empathetic aptitudes and bolstering stress-coping mechanisms, caregivers may forge deeper emotional connections and sustain a more constructive mental equilibrium.”

Line. 114-116

“From a public health perspective, this study could inform policy decisions aiming to improve the overall health and quality of life of caregivers from professionals such as public health nurses.”

  1. Lines 113-128. Why ABI in old age decline with age. Please include some external reference which evidence this statement.

Response: We appreciate the reviewer’s suggestion. We have added refference [26], [27] and added the following text:

Line. 135-139

“The reason for including cases in which the individual with ABI developed ABI at 16 to 64 years old and excluding cases in which ABI occurred in an individual at < 16 or > 64 years old is that adult ABI does not typically progress with age, like pediatric ABI, and does not tend to exhibit decline with age, such as ABI in old age [26].”

Line. 140-143

“A previous study does not the idea that acquired brain injury inevitably causes ongoing neurology decline in the long-term post-injury period because the decline in neurology impairment in old age includes the decline that associated with aging [27].”

  1. In line 118 authors mention that “adult ABI represents a stage that is associated with consistent cognitive and psychosocial function”. But, why is consistent? If every person is different and the psychosocial enviroment could be different too…

Response: We appreciate the reviewer’s suggestion. As your suggestion, difference in individual trajectories in neurology impairment in individuals with traumatic brain injury have been found in previous long-term studies (Corkin, 1989). But this study explained the observed reduction of cognitive capacities late in life and changes in the brain occurring with age. No study has mentioned why adult ABI represents a stage that is associated with consistent cognitive and psychosocial function, and we are sure that this is a topic for future research.

Reference: Corkin S, Rosen TJ, Sullivan EV, Clegg RA. Penetrating head injury in young adulthood exacerbates cognitive decline in later years. J Neurosci . 1989 Nov;9(11):3876-83. doi: 10.1523/JNEUROSCI.09-11-03876.1989.

  1. Lines 258-259, about male caregivers perform statement, please support it with a reference evidence.

Response: We appreciate the reviewer’s suggestion. We provided supporting information for this statement below, so we have corrected as following text:

Line. 335-342

“A previous study suggested that caregiver’s sex may impact psychological adaptation [15]. There is consistent evidence that male caregivers perform better in terms of adaptive status compared with female caregivers as follows. First, males and females tend to have different coping styles, with females taking on caregiving roles with more negative appraisal. In situations involving strong stressors related to caregiving, coping with a stressor using positive appraisal facilitates adaptation to stressors [23,24]. Several previous studies suggest that men experience fewer negative appraisals of caregiving compared with female caregivers [15,41–43].”

  1. In limitations, please, consider the “survival bias” given the age of the individuals in the study.

Response: We appreciate the reviewer’s suggestion. We have added the following text:

Line. 356-359

“Forth, family caregivers who were likely had a lower level of life-change adaptation (e.g., those who had given up caregiving responsibilities because they were getting older) were not included in this study. Thus, this survival bias may have influenced the observed factors.”

  1. In lines 276-278, and accordingly to the manuscript title, why life-change adaptation to aging in care givers was outside the scope of the currently study?.

Response: We appreciate the reviewer’s suggestion. This research question was just to explore the factors contributing to life-change adaptation. The possibility that “caregivers may have difficulty adapting to new life situations or other life changes as they age” was not included in the research question (hypothesis), because it was considered in the results of the current study and was not part of the scope of the current study.

  1. An asterisc is missed en table 1 “Deficits of spatial awareness… 0.041*

Response: We appreciate the reviewer’s suggestion. We have revised table 1.

  1. Abstract: The results need more details.

Response: We appreciate the reviewer’s suggestion. We have revised the Abstract and added the following text:

Line. 19-25

“In total, 312 valid responses and 285 individuals were analyzed for the Poisson regression analysis. The results revealed that life-change adaptation in family caregivers of individuals with ABI would also be related to sex (prevalence ratio [PR]: 0.65; confidence interval [CI]: -0.819 - -0.041), mental health (PR: 2.04; CI: 0.354 - 1.070) as family caregivers’ characteristics and topographical disorientation (PR: 1.51; CI: 0.017 - 0.805) and loss of control over behavior of individuals with ABI (PR: 1.61; CI: 0.116 - 0.830) as the characteristics of individuals with ABI after adjusting for the effects of caregiver’s age, sex, and the duration of the caregiver’s role.”

Round 2

Reviewer 1 Report

We thank the authors for their successful response to all the raised points. There are a few minor concerns that need to be addressed.

a) Could you kindly confirm whether the final sample size was 285? It is not clear why 312 was chosen as the sample size instead of 285. I strongly suggest eliminating this from all the text in this number and presenting all analyses using this number. I kindly request that you delete the sentence regarding the missing date as it has become unnecessary.

b) Change the symbol “-“ in the confidence interval by “;”

c) Please show the sample number used on line 150.

d) In your case, Lilliefors-Corrected Kolmogorov-Smirnov to check the normality distribution of the quantitative variable is more correct than Shapiro Wills

e) Eliminate the text in lines 226-231.

A summary of the results in non-numeric form will be presented in the first paragraph of the discussion.

Author Response

To the comments of Reviewer 1

 Minor changes:

a) Could you kindly confirm whether the final sample size was 285? It is not clear why 312 was chosen as the sample size instead of 285. I strongly suggest eliminating this from all the text in this number and presenting all analyses using this number. I kindly request that you delete the sentence regarding the missing date as it has become unnecessary.

Response: We appreciate the reviewer’s suggestion. The final sample size was 312 and missing data were excluded in the Poisson regression analysis. To perform the correlation analysis with the maximum sample size, 312 participants who met the inclusion and exclusion criteria were considered as the final sample that was eligible for analysis. Thus, Poisson regression analysis was performed with missing values excluded. We have corrected the manuscript to eliminate “285” from the text and all analyses, using the final sample size of “312.” Additionally, we added the text “Missing data were excluded” in the notes for Table 2. The text regarding the missing data was not removed, to help readers to understand this process.

b) Change the symbol “-“ in the confidence interval by “;”

 Response: We appreciate the reviewer’s suggestion. We have revised the relevant text.

c) Please show the sample number used on line 150.

Response: We appreciate the reviewer’s suggestion. We added the following text:

Line 150–151

“Thus, the required sample size was calculated to be 161. The final sample size was 312, which exceeded the pre-study sample size calculation.”

d) In your case, Lilliefors-Corrected Kolmogorov-Smirnov to check the normality distribution of the quantitative variable is more correct than Shapiro Wills.

Response: We appreciate the reviewer’s suggestion. We checked the normality distribution of the quantitative variable using the Lilliefors-Corrected Kolmogorov-Smirnov test. The analysis revealed no change in the results (checked p-value less than 0.05 for all quantitative variables). We have revised the text as follows.

Line 200–202

“The distributions of the quantitative variables were verified using the Lilliefors-corrected Kolmogorov-Smirnov normality test with p < 0.05 for all quantitative variables.”

e) Please, revise. Eliminate the text in lines 226-231. A summary of the results in non-numeric form will be presented in the first paragraph of the discussion.

Response: We agree with the reviewer’s suggestion. In accord with your suggestion, we have removed this summary of the results.

Reviewer 2 Report

Dear authors,

From my point of view, the manuscript has improved in the reviewed version.

Congratulations!

Author Response

We appreciate for the time and energy the reviewer expended. Thank you very much for providing important comments to the reviewer. There is no point-by-point responses to the comments of the reviewer.